# Characteristics of PSA Bounce after Radiotherapy for Prostate Cancer: A Meta-Analysis

**DOI:** 10.3390/cancers12082180

**Published:** 2020-08-05

**Authors:** Narisa Dewi Maulany Darwis, Takahiro Oike, Nobuteru Kubo, Soehartati A Gondhowiardjo, Tatsuya Ohno

**Affiliations:** 1Department of Radiation Oncology, Gunma University Graduate School of Medicine, 3-39-22, Showa-machi, Maebashi, Gunma 371-8511, Japan; m1920021@gunma-u.ac.jp (N.D.M.D.); kubo@gunma-u.ac.jp (N.K.); tohno@gunma-u.ac.jp (T.O.); 2Department of Radiation Oncology, Faculty of Medicine Universitas Indonesia—Dr. Cipto Mangunkusumo National General Hospital, Jl. Diponegoro No. 71, Jakarta Pusat, DKI Jakarta 10430, Indonesia; gondhow@gmail.com; 3Gunma University Heavy Ion Medical Center, 3-39-22, Showa-Machi, Maebashi, Gunma 371-8511, Japan

**Keywords:** prostate cancer, radiotherapy, prostate-specific antigen (PSA), PSA bounce, meta-analysis, meta-regression

## Abstract

The rate and characteristics of prostate-specific antigen (PSA) bounce post-radiotherapy remain unclear. To address this issue, we performed a meta-analysis. Reports of PSA bounce post-radiotherapy with a cutoff of 0.2 ng/mL were searched by using Medline and Web of Science. The primary endpoint was the occurrence rate, and the secondary endpoints were bounce characteristics such as amplitude, time to occurrence, nadir value, and time to nadir. Radiotherapy modality, age, risk classification, androgen deprivation therapy, and the follow-up period were extracted as clinical variables. Meta-analysis and univariate meta-regression were performed with random-effect modeling. Among 290 search-positive studies, 50 reports including 26,258 patients were identified. The rate of bounce was 31%; amplitude was 1.3 ng/mL; time to occurrence was 18 months; nadir value was 0.5 ng/mL; time to nadir was 33 months. Univariate meta-regression analysis showed that radiotherapy modality (29.7%), age (20.2%), and risk classification (12.2%) were the major causes of heterogeneity in the rate of bounce. This is the first meta-analysis of PSA bounce post-radiotherapy. The results are useful for post-radiotherapy surveillance of prostate cancer patients.

## 1. Introduction

Radiotherapy is a definitive treatment for prostate cancer (PCa). Prostate-specific antigen (PSA) is the biomarker used for post-treatment surveillance of PCa patients [1,2]. In curative cases, PSA levels decrease gradually over a period of more than five years after radiotherapy and reach a nadir. In a subset of patients, however, PSA levels fluctuate and show a temporal increase called the PSA bounce [3]. It is difficult to appropriately diagnose PSA increase post-radiotherapy as the bounce; therefore, the PSA increase post-radiotherapy can be the cause of severe anxiety in both PCa patients and clinicians. Misinterpretation may even endanger patients by leading to unnecessary salvage treatment in cases meeting the definition of biochemical failure. PSA bounce can occur in relation to various radiotherapy modalities, including external beam radiotherapy (EBRT), stereotactic body radiotherapy (SBRT), low dose-rate brachytherapy (LDR-BT), and high dose-rate brachytherapy (HDR-BT) [4,5]. As these radiotherapy modalities use different radiation sources, doses, and fractionation, as well as delivery techniques, they can exert different biological effects on the tumor and the prostate. However, the characteristics of PSA bounce in relation to different radiotherapy modalities remain unclear. To address this issue, we performed a meta-analysis of the characteristics of PSA bounce.

## 2. Results

A systematic literature review was performed to identify studies reporting PSA bounce post-radiotherapy (see Materials and Methods for details) (Figure 1). The search identified 50 studies including 26,258 patients, which were included in the analysis (Table 1) [6,7,8,9,10,11,12,13,14,15,16,17,18,19,20,21,22,23,24,25,26,27,28,29,30,31,32,33,34,35,36,37,38,39,40,41,42,43,44,45,46,47,48,49,50,51,52,53,54,55]. The number of studies and patients stratified by modality is summarized in Appendix A. Among the 50 studies, eight were prospective observational studies [14,15,17,32,39,40,41,50] and the others were retrospective observational studies.

A meta-analysis showed that the rate of PSA bounce for all studies was 31% (95% confidence interval (CI), 28–33%) (Figure 2). The bounce rates according to modality were as follows: 34% (95% CI, 30–37%) for LDR-BT, 36% (95% CI, 29–42%) for HDR-BT, 22% (95% CI, 19–25%) for EBRT, 28% (95% CI, 23–32%) for SBRT, 28% (95% CI, 26–31%) for EBRT followed by boost irradiation, and 56% (95% CI, 47–64%) for carbon-ion radiotherapy (Figure 2). For all studies, bounce amplitude was 1.3 ng/mL (95% CI, 1.1–1.4 ng/mL); time to bounce occurrence was 18 months (95% CI, 17–20 months); nadir value was 0.5 ng/mL (95% CI, 0.4–0.6 ng/mL); and time to nadir was 33 months (95% CI, 22–43 months). The results of the meta-analysis stratified by modality are summarized in Table 2, and the original forest plots are shown in Appendix A. Nadir value was higher in bounce-positive patients than in bounce-negative patients for EBRT, SBRT, and CIRT, whereas time to nadir was greater in bounce-positive than in bounce-negative patients regardless of modality (Table 3).

The rate and characteristics of the bounce showed significant heterogeneity among the studies (Table 2). To find the cause of the heterogeneity, we performed univariate meta-regression analysis. Age, radiotherapy modality, use of androgen deprivation therapy (ADT), and risk classification were selected as the covariates for meta-regression based on previous studies reporting that these factors affect the bounce kinetics [4,5]. The heterogeneity in the bounce rate was attributed to modality (29.7%), age (20.2%), and risk classification (12.2%) (Figure 3A,B, Table 3). Regarding bounce amplitude, age was a significant cause of heterogeneity (Figure 3C, Table 3). For time to bounce occurrence, modality was a significant cause of heterogeneity (Table 4).

## 3. Discussion

The strength of this study is that this is the first meta-analysis to investigate the characteristics of PSA bounce post-radiotherapy. We report the rate, amplitude, nadir, and time course of the bounce for different modalities including brachytherapy, EBRT, SBRT, and CIRT. We also report that the bounce occurs more frequently and with greater amplitude in brachytherapy than in EBRT, and a younger age is associated with a higher incidence and greater amplitude of the bounce. These findings have been extensively reported in mono-institutional studies, e.g., the large-scale study by Romesser [46], which were validated here for the first time by meta-analysis. From this standpoint, the results of the present study are useful for post-radiotherapy surveillance of prostate cancer patients to help oncologists and patients interpret temporal PSA increases post-treatment.

The limitations of this study, on the other hand, are the following. First, the studies analyzed were extremely heterogeneous regarding clinical factors such as dose, fractionation, bounce rate according to ADT usage, and risk classification, which was difficult to control in a meta-analysis design. In particular, the ADT strategy (i.e., the presence or absence of adjuvant or neoadjuvant use) should have affected post-radiotherapy PSA kinetics to a large extent, which was difficult to adjust by study design. Second, we were not able to analyze the PSA kinetics post-radiotherapy stratified by bounce positivity except for nadir and time to nadir. This was because extraction of the corresponding data from the original articles was technically impossible; i.e., the original articles did not contain the PSA kinetics data linked to specific clinical variables (e.g., age and risk) in a form that we can compute in the meta-analysis. Third, we were unable to perform multivariate meta-regression analysis because of the small number of studies. Fourth, most of the studies included had a retrospective design, and no randomized studies were identified. Finally, studies on particle therapy were rarely identified (i.e., one study on CIRT and no studies on proton therapy).

The molecular mechanisms underlying PSA bounce remain to be elucidated. Studies have shown that PSA is released from both tumor tissues and the normal prostate glands after irradiation [48]. Radiation-induced antitumor immunity may contribute to the release of PSA from tumor tissues. For example, Yamamoto et al. reported intra-tumoral infiltration of CD3- and CD8-positive lymphocytes in bounce-positive patients [56]. In the present meta-analysis, the bounce was more prevalent after brachytherapy and SBRT than after EBRT. In addition, the bounce rate for CIRT was strikingly high, although only one study was analyzed. These findings may be explained by the highly concentrated dose delivery by brachytherapy, SBRT, and CIRT compared with that of EBRT. Evidence suggests that a high, single-fractionated dose induces antitumor immunity efficiently [57], partially by promoting DNA damage response signaling [58]. In addition, the properties of carbon ions as high linear energy transfer radiation to efficiently induce antitumor immunity (e.g., induction of HMGB1 [59], OX40L, CD40, ICAM-1, and MHC-1, and suppression of PD-L1 [60]) might contribute to the high bounce rate for CIRT. Another possible explanation for the higher bounce rate associated with brachytherapy, SBRT, and CIRT is that the highly concentrated doses delivered by these modalities destroy the normal prostate glands more efficiently. Kirilova et al. showed an increase in metabolism indicative of inflammation in the normal prostate gland of patients experiencing bounce, which supports this notion [61].

In addition to modality, the meta-regression results indicated that younger age is associated with greater bounce occurrence and amplitude. This is consistent with the findings of the systematic literature review, in which 29 of the 50 papers analyzed identify younger age as a predictor of bounce. Yamamoto et al. suggested that this may be related to the higher immunocompetency in younger patients [56]. Further research is warranted to elucidate immunologic responses of PCa and the prostate glands after radiotherapy.

## 4. Materials and Methods

### 4.1. Endpoint Definition

The primary endpoint of this study was the rate of PSA bounce. Secondary endpoints included the characteristics of bounce, i.e., bounce amplitude, time to occurrence, nadir value, and time to nadir. Definitions of these endpoints are listed in Appendix A.

### 4.2. Inclusion and Exclusion Criteria

The inclusion criteria were as follows: (i) an original clinical study reporting on radiotherapy for PCa; (ii) available rate of PSA bounce; and (iii) bounce defined as an increase in PSA over a cutoff of 0.2 ng/mL followed by a spontaneous decrease to or below the pre-bounce nadir [19]. The exclusion criteria were as follows: (i) manuscript written in languages other than English; (ii) full manuscript not available; (iii) subgroup analysis of a given reported cohort; (iv) follow-up shorter than 24 months.

### 4.3. Study Selection

A systematic literature search based on preferred reporting items for systematic reviews and meta-analyses (PRISMA) guidelines [62] was performed on 20 March 2020, using two databases, Medline and Web of Science. The search strategy and population-intervention-comparison-outcome metrics [63] are described in Appendix A, respectively. The search results were combined using the bibliographic management software Mendeley Desktop version 1.19.4 (Mendeley, London, UK), and duplicates were eliminated. Two investigators (N.D.M.D. and T.Oi.) independently reviewed all records in the following three steps. In step 1, the titles of all records were reviewed to detect potentially relevant records. In step 2, the abstracts of all records that passed step 1 were reviewed to detect potentially relevant records. In step 3, the entire manuscripts of all records that passed step 2 were examined if they contained extractable data for the primary endpoint.

### 4.4. Data Extraction

From the studies identified in Section 4.3, two investigators (N.D.M.D. and T.Oi.) independently extracted the following data: primary and secondary endpoints, radiotherapy modality, age, risk classification [64], the use of ADT, and follow-up period.

### 4.5. Quality Assessment

Two investigators (N.D.M.D. and T.Oi.) independently confirmed that the methodological quality of the included studies was adequate based on the Quality Assessment Tool for Case Series Studies published by the National Heart, Lung, and Blood Institute-National Institute of Health, U.S. [65]. For Section 4.3, Section 4.4, and Section 4.5, decisions were made based on discussion by the two investigators to resolve disagreements on the review results.

### 4.6. Statistical Analysis

Radiotherapy modalities were classified into six groups as follows: iLDR-BT (^103^Pd, ^125^I, or ^131^Cs), HDR-BT (^192^Ir), EBRT (three-dimensional conformal radiotherapy or intensity-modulated radiation therapy), SBRT (using CyberKnife or linac), EBRT+boost (using LDR-BT, HDR-BT, or SBRT), and CIRT. Meta-analysis of bounce (binomial data) was performed using *metaprop*, a command of Stata (MP 13, StataCorp, College Station, TX, USA) [66]. Meta-analysis of the characteristics of bounce (continuous variables) was performed using *metan*, a Stata command. For the datasets that lacked the mean and standard deviation to be pooled, these values were estimated from the sample size, median, range, and/or interquartile range, as reported previously [67]. A random-effects model was used considering a high extent of inter-study heterogeneity examined using *X^2^* and *I^2^* statistics [68]. Meta-regression was performed to analyze the effect of clinical factors on inter-study heterogeneity in effect size using *metareg*, a Stata command [69]. To construct the *metareg* command for bounce rate, logit prevalence and its standard error were used [70,71]; for the remaining PSA kinetics outcomes, mean and standard error were used [72]. Results with a *p*-value < 0.05 were interpreted as significant.

## 5. Conclusions

This is the first study to report the results of meta-analysis and meta-regression of PSA bounce post-radiotherapy. Meta-analysis of 50 studies including 26,258 patients showed that the rate of PSA bounce for all studies was 31% (95% CI, 28–33%); bounce amplitude was 1.3 ng/mL (95% CI, 1.1–1.4 ng/mL); time to bounce occurrence was 18 months (95% CI, 17–20 months); nadir value was 0.5 ng/mL (95% CI, 0.4–0.6 ng/mL); and time to nadir was 33 months (95% CI, 22–43 months). The bounce occurred more frequently and with greater amplitude in brachytherapy than in EBRT. Univariate meta-regression showed that younger age is associated with a higher incidence and greater amplitude of bounce. These data will be useful for post-radiotherapy surveillance of PCa patients to help oncologists and patients interpret temporal PSA increases post-treatment.

## Figures and Tables

**Figure 1 cancers-12-02180-f001:**
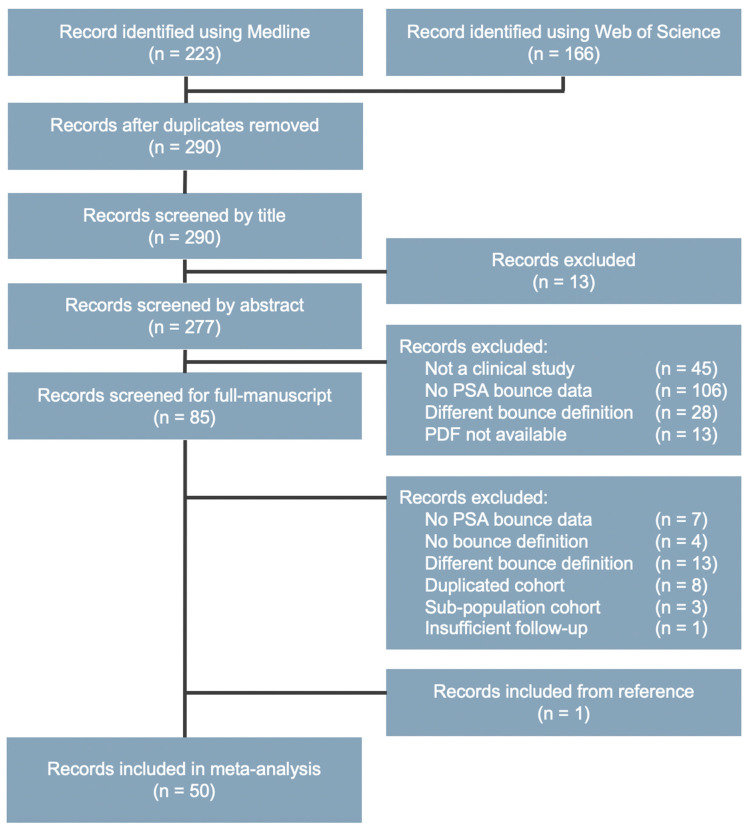
Preferred reporting items for systematic reviews and meta-analyses (PRISMA) flow diagram of the literature review for prostate-specific antigen (PSA) bounce after radiotherapy.

**Figure 2 cancers-12-02180-f002:**
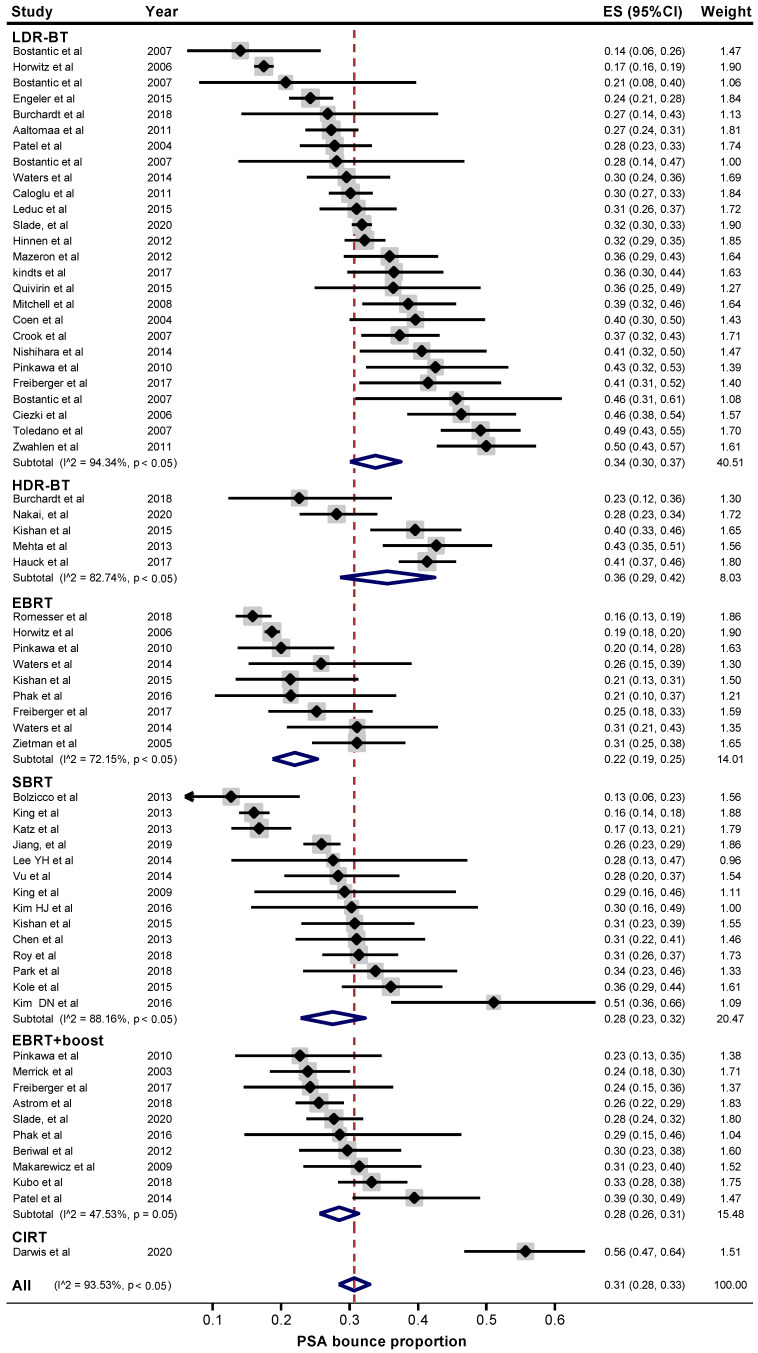
Meta-analysis of the rate of prostate-specific antigen (PSA) bounce after radiotherapy. ES, effect size; CI, confidence interval; LDR-BT, low dose-rate brachytherapy; HDR-BT, high dose-rate brachytherapy; EBRT, external beam radiotherapy; SBRT, stereotactic body radiotherapy; CIRT, carbon ion radiotherapy.

**Figure 3 cancers-12-02180-f003:**
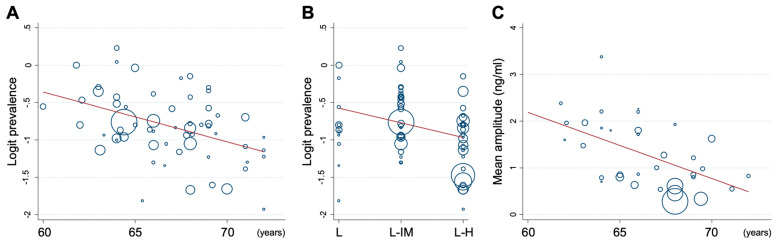
Univariate meta-regression of heterogeneity in the rate of bounce by age (**A**) or by risk group (**B**), and that in bounce amplitude by age (**C**). L, low risk; IM, intermediate risk; H, high risk.

**Table 1 cancers-12-02180-t001:** Papers that report PSA bounce after radiotherapy included in the meta-analysis.

Author	Year	*n*	Modality	Age	Risk Group	ADT	Follow Up (M)	Bounce (%)	Amplitude (ng/mL)	Time to Bounce (M)	Nadir (ng/mL)	Time to Nadir (M)	Reference
Merrick et al.	2002	218	EBRT+LDR-BT	66 ± 7	L, I	No	46 ± 14	23.9	0.9 (0.3–3.0)	19 ± 9	NA	NA	[6]
Patel et al.	2004	295	LDR-BT	NA	L, I	Yes, partly	38 (24–68)	28.0	0.5 (0.2–4.1)	19 (8–40)	NA	NA	[7]
Coen et al.	2004	101	LDR-BT	NA	L, I	No	54 (38–86)	39.6	0.6 (0.2–7.5)	18 (7–71)	NA	NA	[8]
Zietman et al.	2005	190	EBRT	NA	L, I, H	Yes, all	60 (40–75)	39.0	0.9 (0.5–1.8)	28 (17–42)	NA	NA	[9]
Ciezki et al.	2006	162	LDR-BT	68 (45–83)	L, I, H	Yes, partly	73	46.3	NA	15 (2–57)	NA	NA	[10]
Horwitz et al.	2006	4839	EBRT	NA	L, I, H	No	75	18.6	NA	NA	NA	NA	[11]
		2693	LDR-BT	NA	L, I, H	No	60	17.5	NA	NA	NA	NA	
Toledano et al.	2007	295	LDR-BT	60–65	L, I	Yes, partly	40 (9–66)	49.0	0.8, mean (0.1–4.1)	19, mean (6–58)	NA	NA	[12]
Bostantic et al.	2007	57	LDR-BT	65 ± 6	L	No	62 ± 10	14.0	0.4	18 ± 9	NA	NA	[13]
		46	LDR-BT	63 ± 7	L	No	64 ± 12	45.7	0.4	22 ± 11	NA	NA	
		29	LDR-BT	66 ± 6	L	Yes, all	67 ± 12	20.7	0.4	6 ± 6	NA	NA	
		32	LDR-BT	67 ± 5	L	Yes, all	62 ± 12	28.1	0.4	18 ± 8	NA	NA	
Crook et al.	2007	292	LDR-BT	64 (45–80)	L, I	No	44 (8–81)	40.0	0.7 (0.2–11.7)	15 (3–29)	0.05 (0.01–0.20)	40	[14]
Mitchell et al.	2008	205	LDR-BT	62, mean (43–75)	L, I	No	45 (24–85)	37.0	0.9 (0.2–5.8)	14 (1–40)	NA	NA	[15]
Makarewicz et al.	2009	121	EBRT+HDR-BT	68 (47–78)	L, I	No	81 (60–106)	31.0	0.2, mean (0.2–0.7)	14 (7–26)	0.8 (0.01–2.1)	NA	[16]
King et al.	2009	41	SBRT	66 (48–83)	L	No	33 (6–45)	29.0	0.3 (0.2–2.4)	18 (12–33)	0.3 (0.03–2.6)	NA	[17]
Pinkawa et al.	2010	135	EBRT	71 (52–83)	L, I, H	Yes, partly	67 (9–97)	20.0	NA	NA	NA	NA	[18]
		66	EBRT+HDR-BT	72 (63–81)	L, I, H	Yes, partly	75 (7–98)	23.0	NA	NA	NA	NA	
		94	LDR-BT	69 (49–81)	L, I	Yes, partly	76 (8–101)	42.0	NA	NA	NA	NA	
Caloglu et al.	2011	820	LDR-BT	68 (45–87)	L, I, H	Yes, partly	58 (36–123)	30.1	NA	17 (2–68)	NA	NA	[19]
Zwahlen et al.	2011	194	LDR-BT	61 (47–75)	L	No	60 (23–109)	50.0	0.5 (0.2–8.3)	14 (0–70)	0.1 (0.0–3.5)	NA	[20]
Aaltomaa et al.	2011	535	LDR-BT	64 (42–80)	L, I, H	Yes, partly	69 (15–131)	27.4	NA	NA	NA	NA	[21]
Beriwal et al.	2012	155	EBRT+LDR-BT	65 ± 7	L, I, H	Yes, partly	36 (24–60)	29.7	0.6, mean (0.2–2.3)	12, mean (6–36)	NA	NA	[22]
Hinnen et al.	2012	975	LDR-BT	66 ± 6	L, I, H	Yes, partly	78 (27–215)	32.0	1.7 (1.0–2.8, IQR)	19 (12–24, IQR)	NA	12 (6–15)	[23]
Mazeron et al.	2012	198	LDR-BT	67 (49–80)	L, I	No	63 (36–119)	35.9	1.0 ± 1.0	18 ± 9	NA	NA	[24]
Bolzicco et al.	2013	71	SBRT	72 (52–82)	L, I, H	Yes, partly	36 (6–76)	12.6	NA	23 (18–30)	0.4	36	[25]
Chen et al.	2013	100	SBRT	69 (48–90)	L, I, H	Yes, partly	27 (16–42)	31.0	0.5 (0.2–2.2)	15 (3–21)	0.4 (0.1–1.9)	24	[26]
Katz et al.	2013	304	SBRT	69, mean (45–88)	L, I, H	Yes, partly	60 (8–78)	17.0	0.5	30	0.1	60	[27]
King et al.	2013	1100	SBRT	70 (44–91)	L, I, H	Yes, partly	36	16.0	0.5 (0.2–5.29)	NA	NA	NA	[28]
Mehta et al.	2013	157	HDR-BT	63 (42–90)	L, I	Yes, partly	55	43.0	0.6 (0.2–4.5)	13 (0.6–64)	NA	NA	[29]
Lee et al.	2014	29	SBRT	72 (50-86)	L, I, H	Yes, partly	41 (12–69)	28.0	0.6 (0.3–1.5)	9	0.3 (0.003–1.7)	23	[30]
Nishihara et al.	2014	116	LDR-BT	66 (51–80)	L, I	No	42 (18–77)	40.5	0.4 (0.2–5.6)	17 (8–36)	NA	NA	[31]
Vu et al.	2014	120	SBRT	68 (47–88)	L, I, H	Yes, partly	24 (18–78)	28.0	0.5	9	NA	NA	[32]
Patel et al.	2014	114	EBRT+HDR-BT	68 (48–79)	L, I	No	66 (24–124)	39.0	0.4 (0.2–6.6)	16 (3–76)	0.1 (0.01–1.7)	53 (8–118)	[33]
Waters et al.	2014	74	EBRT, hopo	68 ± 5	L	No	36, min	31.1	0.6	NA	NA	NA	[34]
		58	EBRT	66 ± 5	L	No	36, min	20.7	0.3	NA	NA	NA	
		230	LDR-BT	64 ± 6	L	No	36, min	29.6	0.6	NA	NA	NA	
Kole et al.	2015	175	SBRT	69 (48–85)	L, I, H	No	36	36.2	NA	15 (1–42)	0.3 (0.02–1.8)	30 (3–48)	[35]
Kishan et al.	2015	130	SBRT	69 (44–87)	L, I	No	40 (12–93)	30.8	0.5 (0.2–3.6)	14 (3–43)	NA	NA	[36]
	2015	220	HDR-BT	64 (43–84)	L, I	No	49 (12–94)	39.5	0.6 (0.2–7.1)	10 (3–63)	NA	NA	
	2015	89	EBRT	66 (52–85)	L, I	No	27 (12–90)	21.3	0.5 (0.2–7.6)	13 (3–66)	NA	NA	
Leduc et al.	2015	274	LDR-BT	62 (45–76)	L	Yes, partly	50 (24–126)	31.0	1.0 (0.2–12.4)	12 (6–37)	NA	NA	[37]
Quivirin et al.	2015	66	LDR-BT	64 ± 5	L	No	35 (13–72)	36.4	1.8 ± 1.6	12 ± 6	NA	NA	[38]
Engeler et al.	2015	713	LDR-BT	63 (42–82)	L, I, H	Yes, partly	41 (24–132)	24.3	0.7 (0.2–6.1)	12 (6–33)	NA	NA	[39]
Kim et al.	2016	33	SBRT	67 (56–72)	L, I	No	51 (6–71)	30.3	0.2 (0.2–1.3)	10 (6–12)	0.2	33	[40]
Kim et al.	2016	47	SBRT	64 (52–82)	L, I	No	42 (36–78)	51.0	0.5 (0.2–6.2)	9 (3–36)	NA	36 (11, SD)	[41]
Phak et al.	2016	35	EBRT+SBRT	69, mean (60–78)	L, I	No	52 (14–74)	28.6	0.2 (0.2–0.5)	11 (6–25)	0.2 (0.04–1.4)	32 (12–51)	[42]
		42	EBRT	71, mean (61–79)	L, I	No	52 (14–74)	21.4	0.3 (0.2–1.2)	15 (6–30)	0.3 (0.04–1.8)	25 (9–58)	
Freiberger et al.	2017	94	LDR-BT	69 (49–81)	L, I	Yes, partly	108	42.0	NA	NA	0.05, mean	32, mean	[43]
		66	EBRT+HDR-BT	72 (63–81)	L, I, H	Yes, partly	108	24.0	NA	NA	0.1, mean	31, mean	
		135	EBRT	71 (52–83)	L, I, H	Yes, partly	108	25.0	NA	NA	0.5, mean	19, mean	
Hauck et al.	2017	554	HDR-BT	63 (40–83)	L, I, H	Yes, partly	44 (12–162)	43.2	NA	11, mean	0.2	NA	[44]
Kindts et al.	2017	192	LDR-BT	60 (50–65)	L, I	Yes, partly	66	36.0	0.6, mean	18, mean	NA	NA	[45]
Romesser et al.	2017	776	EBRT	61-72, IQR	L, I, H	Yes, partly	110 (83–134)	15.9	0.3 (0.2–0.7, IQR)	24 (16–38, IQR)	NA	NA	[46]
Park et al.	2018	74	SBRT	69 (47–81)	L, I, H	No	63 (12–109)	35.2	0.5 (0.2–2.6)	11 (2–38)	0.1 (0.01–2.6)	47 (1–85)	[47]
Astrom et al.	2018	623	EBRT+HDR-BT	66 (47–79)	L, I, H	Yes, partly	132 (2–266)	26.0	1.5 (0.3–12.0)	15 (3–103)	NA	NA	[48]
Burchardt et al.	2018	41	LDR-BT	64 ± 7	L, I	Yes, partly	37 ± 8	26.8	0.7 ± 1.1	18 ± 6	0.5 ± 1.1	23 ± 14	[49]
		53	HDR-BT	67 ± 7	L, I	Yes, partly	33 ± 9	22.6	0.8 ± 0.5	10 ± 4	0.2 ± 0.4	19 ± 14	
Kubo et al.	2018	352	EBRT+LDR-BT	69 (49–82)	L, I, H	Yes, partly	82 (12–157)	33.2	NA	20 (3–55)	NA	NA	[50]
Roy et al.	2019	287	SBRT	69 (49–82, IQR)	L, I	Yes, partly	60 (46–106)	31.1	0.6 (0.35–1.61, IQR)	17 (11–25, IQR)	NA	NA	[51]
Jiang et al.	2019	1062	SBRT	68 (63–73, IQR)	L, I	No	66 (36–60, IQR)	26.0	0.5 (0.3–1.0, IQR)	18 (12–31, IQR)	0.2 (0.1–0.3, IQR)	40 (24–66, IQR)	[52]
Darwis et al.	2020	131	Carbon ions	64, mean (48–80)	L, I	No	60 (39–60)	55.7	0.7 ± 1.0	15 ± 11	0.5 ± 0.3	42 (9–60)	[53]
Nakai et al.	2020	256	HDR-BT	67 ± 6	L, I	No	91 ± 23	32.3	NA	19 ± 23	NA	NA	[54]
Slade et al.	2020	4004	LDR-BT	64 ± 6	L, I	No	120	31.8	NA	NA	NA	NA	[55]
		473	EBRT	64 ± 6	L, I	No	120	27.7	NA	NA	NA	NA	

PSA, prostate-specific antigen; EBRT, external beam radiotherapy; LDR-BT, low dose-rate brachytherapy; HDR-BT, high dose-rate brachytherapy; SBRT, stereotactic body radiotherapy; CIRT, carbon ion radiotherapy; NA, not assessable; IQR, interquartile range; L, low risk; I, intermediate risk; H, high risk; ADT, androgen deprivation therapy; M, months. Age, follow-up, and bounce outcomes are shown as mean ± standard deviation or in median (range) unless otherwise stated.

**Table 2 cancers-12-02180-t002:** Summary of the results of the meta-analysis of PSA bounce characteristics.

Modality	Rate of Bounce (%)	Amplitude (ng/mL)	Time to Occurrence (M)	Nadir (ng/mL)	Time to Nadir (M)
LDR-BT	34 (30–37)	1.7 (1.3–2.0)	18 (17–20)	0.5 (−0.1–1.1)	23 (19–28)
HDR-BT	36 (29–42)	1.4 (0.7–2.2)	18 (12–25)	0.2 (0.09–0.3)	19 (15–23)
EBRT	22 (19–25)	0.8 (0.4–1.2)	24 (20–29)	0.6 (0.5–0.7)	29 (25–32)
SBRT	28 (23–32)	1.0 (0.7–1.2)	17 (14–20)	0.6 (0.3–0.8)	38 (26–51)
EBRT + boost	28 (26–31)	1.0 (0.7–1.4)	18 (14–22)	0.6 (0.4–0.8)	44 (19–70)
CIRT	56 (47–64)	0.7 (0.5–1.0)	15 (12–17)	0.5 (0.4–0.6)	42 (40–44)
Pooled ES	31 (28–33)	1.3 (1.1–1.4)	18 (17–20)	0.5 (0.4–0.6)	35 (28–42)
*I*^2^, *p* values	93.5% (*p* < 0.05)	98.3% (*p* < 0.05)	95.7% (*p* < 0.05)	99.5% (*p* < 0.05)	98.4% (*p* < 0.05)

PSA, prostate-specific antigen; LDR-BT, low dose-rate brachytherapy; HDR-BT, high dose-rate brachytherapy; EBRT, external beam radiotherapy; SBRT, stereotactic body radiotherapy; CIRT, carbon ion radiotherapy; ES, effect size; M, months. Data are means (95% confidence interval).

**Table 3 cancers-12-02180-t003:** Summary of the results of the meta-analysis of PSA nadir and time to nadir stratified by bounce occurrence.

Modality	Nadir (ng/mL)		Time to Nadir (M)	
	Bounce	No bounce	Bounce	No bounce
LDR-BT	NA	NA	NA	NA
HDR-BT	NA	NA	NA	NA
EBRT	0.7 (0.7–0.8)	0.5 (0.5–0.5)	42 (40–43)	29 (28–29)
SBRT	0.6 (0.5–0.7)	0.3 (0.3–0.4)	NA	NA
EBRT + Boost	0.3 (0.2–0.4)	0.5 (0.4–0.6)	64 (58–70)	54 (48–60)
CIRT	0.6 (0.5–0.7)	0.4 (0.3–0.5)	48 (45–50)	36 (33–40)
Pooled effect size	0.6 (0.3–0.8)	0.5 (0.4–0.6)	50 (42–59)	39 (27–51)
*I*^2^, *p* values	98.5% (*p* < 0.05)	95.1% (*p* < 0.05)	96.6% (*p* < 0.05)	97.6% (*p* < 0.05)

PSA, prostate-specific antigen; LDR-BT, low dose-rate brachytherapy; HDR-BT, high dose-rate brachytherapy; EBRT, external beam radiotherapy; SBRT, stereotactic body radiotherapy; CIRT, carbon ion radiotherapy; NA, not assessible; M, months. Data are means (95% confidence interval).

**Table 4 cancers-12-02180-t004:** Univariate meta-regression for the proportion and characteristics of bounce.

Covariates	Rate of Bounce (*n* = 65)			Amplitude (*n* = 37)			Time to Occurrence (*n* = 45)			Nadir (*n* = 15)			Time to Nadir (*n* = 9)		
	Coefficient	*p*	R^2^ (%)	Coefficient	*p*	R^2^ (%)	Coefficient	*p*	R^2^ (%)	Coefficient	*p*	R^2^ (%)	Coefficient	*p*	R^2^ (%)
Age	−0.07 (−0.10 to −0.03)	<0.01	20.2	−0.14 (−0.22 to −0.06)	<0.01	28.1	0.30 (−0.27 to 0.87)	0.33	1.1	−0.01 (−0.06 to 0.05)	0.78	0.0	−0.32 (−4.82 to 4.18)	0.87	0.0
Modality															
LDR-BT	−0.08 (−0.46 to 0.30)	0.66	29.7	0.25 (−0.61 to 1.10)	0.56	15.4	−5.57 (−11.32 to 0.18)	0.05	5.0	−0.05 (−0.79 to 0.68)	0.88	0.0	−18.96 (−78.66 to 40.74)	0.38	0.0
HDR-BT	NA	NA		NA	NA		NA	NA		NA	NA		NA	NA	
EBRT	−0.63 (−1.07 to −0.19)	<0.01		−0.53 (−1.56 to 0.50)	0.30		NA	NA		0.11 (−0.79 to 1.01)	0.78		−13.31 (−72.86 to 46.24)	0.52	
SBRT	−0.38 (−0.79 to 0.03)	0.07		−0.48 (−1.36 to 0.40)	0.27		−7.24 (−13.36 to –1.11)	0.02		0.07 (−0.61 to 0.75)	0.82		−3.90 (−52.43 to 44.62)	0.81	
EBRT + boost	−0.32 (−0.75 to 0.11)	0.14		−0.41 (−1.35 to 0.53)	0.38		−5.92 (−12.39 to 0.56)	0.07		0.07 (−0.65 to 0.80)	0.82		2.23 (−49.31 to 53.77)	0.89	
CIRT	0.83 (−0.26 to 1.68)	0.05		−0.70 (−2.24 to 0.83)	0.35		−9.50 (−20.68 to 1.67)	0.09		NA	NA		NA	NA	
ADT	−0.14 (−0.34 to 0.06)	0.17		−0.20 (−0.63 to 0.22)	0.34	0.8	0.32 (−2.18 to 2.82)	0.79	0.0	−0.07 (−0.41 to 0.27)	0.66	0.0	−18.10 (−37.28 to 1.08)	0.06	33.8
Risk group	−0.20 (−0.35 to −0.04)	0.01	12.2	−0.23 (−0.62 to 0.15)	0.22	1.3	1.42 (−0.78 to 3.61)	0.20	0.0	0.02 (−0.21 to 0.24)	0.88	0.0	0.94 (−24.00 to 25.89)	0.93	0.0

LDR-BT, low dose-rate brachytherapy; HDR-BT, high dose-rate brachytherapy; EBRT, external beam radiotherapy; SBRT, stereotactic body radiotherapy; CIRT, carbon ion radiotherapy; ADT, androgen deprivation therapy. NA, not assessible due to collinearity. Data are means (95% confidence interval).

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
