# Peer review of "Characteristics of PSA Bounce after Radiotherapy for Prostate Cancer: A Meta-Analysis"

_cancers, 2020, doi:10.3390/cancers12082180_

Round 1

Reviewer 1 Report

This is an interesting meta-analysis on PSA "bounce" after radio-therapy. However, the analysis came out a little bit short. It is known from a 2018 analysis from the Memorial Sloan-Kettering Cancer Center (New York) that

  • 16 percent of patients had a bounce.
  • The bounce occurred after an average (median) of 24.6 months.
  • It was an average (median) of 0.37 ng/ml over the previous nadir.

Bounces were more likely to occur in patients who:

  • Were younger (with a link)
  • Had a lower Gleason score
  • Had a lower T stage
  • Received a higher radiation dose

At 8-years follow-up, they reported that bounces were associated with:

  • Greater time to reach ultimate PSA nadir (43 vs 26 months)
  • Lower risk of PSA relapse (9 vs 29 percent)
  • Decreased risk of metastases (1 vs 10 percent)
  • Decreased prostate-specific mortality (0 vs 3 percent)
  • Decreased overall mortality (6 vs 11 percent)

So, for this meta-analysis submitted here, it will be very interesting to know, if possible to extract from the selected publications, the amplitude, time to occurrence, nadir value, and time to nadir for both patients with or without "bounce" post-therapy instead of a lumped average in the current analysis, and under each uni-variable (modality, age, and risk), if possible. Please try that.

Author Response

Reviewer 1:

This is an interesting meta-analysis on PSA "bounce" after radiotherapy.

Response:

We sincerely thank the reviewer for evaluating the manuscript and for the encouraging comments. In accordance with the suggestions, the manuscript was revised as follows.

However, the analysis came out a little bit short. It is known from a 2018 analysis from the Memorial Sloan-Kettering Cancer Center (New York) that

  • 16 percent of patients had a bounce.
  • The bounce occurred after an average (median) of 24.6 months.
  • It was an average (median) of 0.37 ng/ml over the previous nadir.

Bounces were more likely to occur in patients who:

  • Were younger (with a link)
  • Had a lower Gleason score
  • Had a lower T stage
  • Received a higher radiation dose

At 8-years follow-up, they reported that bounces were associated with:

  • Greater time to reach ultimate PSA nadir (43 vs 26 months)
  • Lower risk of PSA relapse (9 vs 29 percent)
  • Decreased risk of metastases (1 vs 10 percent)
  • Decreased prostate-specific mortality (0 vs 3 percent)
  • Decreased overall mortality (6 vs 11 percent)

Response:

We sincerely thank the reviewer for introducing the results of the study by Romesser et al. (Int J Radiat Oncol Biol Phys, 2018; 100: 59-67). We acknowledged this landmark study in Discussion section (lines 114116).

So, for this meta-analysis submitted here, it will be very interesting to know, if possible to extract from the selected publications, the amplitude, time to occurrence, nadir value, and time to nadir for both patients with or without "bounce" post-therapy instead of a lumped average in the current analysis, and under each uni-variable (modality, age, and risk), if possible. Please try that.

Response:

We sincerely thank the reviewer for the insightful comments. In accordance with the suggestion, we tried to perform further analysis. As a result, we successfully extracted and pooled the data from the selected publications pertaining to nadir value and time to nadir stratified by radiotherapy modality; nadir value was higher in bounce-positive patients than in bounce-negative patients for EBRT, SBRT, and CIRT, whereas time to nadir was greater in bounce-positive than in bounce-negative patients regardless of modality. The data were provided as Table 3 and described in the lines 7375. Meanwhile, unfortunately, we were not able to analyze the PSA kinetics post-radiotherapy stratified by bounce positivity except for nadir and time to nadir. This was because extraction of the corresponding data from the original articles was technically impossible; i.e., the original articles did not contain the PSA kinetics data linked to specific clinical variables (e.g., age and risk) in a form that we can compute in the meta-analysis. Therefore, we discussed this issue as the limitation of this study in the lines 124128. We hope your understanding and thank you again for the valuable comment that improved the quality of our manuscript.

Reviewer 2 Report

This manuscript investigates the characteristics of PSA bounce postradiotherapy in prostate cancer using a meta-analysis to reporting rate, amplitude, nadir, and time course of the bounce for different radiotherapy modalities. The results obtained suggest that the bounce occurs more frequently in brachytherapy, and a younger age is associated with a higher incidence and greater amplitude of the bounce. These data could be useful for oncologists to manage post-radiotherapy surveillance of prostate cancer patients.  

Overall, this research is well written, and the content of this manuscript is of major interest. Nevertheless, the following issues (minors) need to be addressed:

- Lines 21-24: In my opinion, the abstract should be the resume of your results. I would avoid presenting too many numbers in the abstract

- Line 32: Please use the abbreviation PC or PCa for prostate cancer

- Line 36: This sentence is hard to follow. Please rephrase

- Line 131: I appreciate the self-criticism of the authors that who admit the limitations of their study. However, I would not finish the discussion with “this study had several limitations”. You can move these sentences above in the discussion.

Author Response

Reviewer 2:

This manuscript investigates the characteristics of PSA bounce postradiotherapy in prostate cancer using a meta-analysis to reporting rate, amplitude, nadir, and time course of the bounce for different radiotherapy modalities. The results obtained suggest that the bounce occurs more frequently in brachytherapy, and a younger age is associated with a higher incidence and greater amplitude of the bounce. These data could be useful for oncologists to manage post-radiotherapy surveillance of prostate cancer patients. Overall, this research is well written, and the content of this manuscript is of major interest. Nevertheless, the following issues (minors) need to be addressed:

Response:

We sincerely thank the reviewer for evaluating the manuscript and for the encouraging comments. In accordance with the suggestions, the manuscript was revised as follows.

Lines 21-24: In my opinion, the abstract should be the resume of your results. I would avoid presenting too many numbers in the abstract.

Response:

In accordance with the suggestion, the presentation of the numbers in that part was decreased by 66% (lines 2122). We sincerely thank the reviewer for the valuable suggestion that improved the quality of the abstract.

Line 32: Please use the abbreviation PC or PCa for prostate cancer

Response:

In accordance with the suggestion, all "prostate cancer" in the main text was thoroughly changed to PCa (lines 31, 32, 36, 153, 162, and 210). We sincerely thank the reviewer for the suggestion.

Line 36: This sentence is hard to follow. Please rephrase.

Response:

In according with the suggestion, the sentence was rephrased as follows: "It is difficult to appropriately diagnose PSA increase post-radiotherapy as the bounce; therefore, the PSA increase post-radiotherapy can be the cause of severe anxiety in both PCa patients and clinicians"(lines 34-36). We sincerely thank the reviewer for the comment.

Line 131: I appreciate the self-criticism of the authors that who admit the limitations of their study. However, I would not finish the discussion with “this study had several limitations”. You can move these sentences above in the discussion.

Response:

We sincerely thank the reviewer for the insightful comment. In accordance with the suggestion, the limitation paragraph was re-located upward (lines 119-131). To keep alignment of the entire Discussion section, the following sentences were re-phrased (lines 110, 119, and 153-154).

Round 2

Reviewer 1 Report

The authors did the additional extraction. Since PSA kinetics are not available, that is fine...